# A Comparison of Doxycycline and Amoxicillin Containing Quadruple Eradication Therapy for Treating *Helicobacter pylori*-Infected Duodenal Ulcers: A Multicenter, Opened, Randomized Controlled Trial in China

**DOI:** 10.3390/pathogens11121549

**Published:** 2022-12-16

**Authors:** Jingshu Chi, Canxia Xu, Xiaoming Liu, Hao Wu, Xiaoran Xie, Peng Liu, Huan Li, Guiying Zhang, Meihua Xu, Chaomin Li, Chunlian Wang, Fengqian Song, Ming Yang, Jie Wu

**Affiliations:** 1Department of Gastroenterology, Third Xiangya Hospital of Central South University, Changsha 410013, China; 2Xiangya Changde Hospital, Changde 415000, China; 3Department of Gastroenterology, Xiangya Hospital of Central South University, Changsha 410008, China; 4Shaoyang Central Hospital, Shaoyang 422000, China; 5Department of Gastroenterology, Second Xiangya Hospital of Central South University, Changsha 410011, China; 6Loudi Central Hospital, Loudi 417000, China; 7The Affiliated Hospital of Yongzhou Vocational Technical College, Yongzhou 425006, China

**Keywords:** *Helicobacter pylori*, antibiotic resistance, doxycycline, eradication, safety, penicillin allergy

## Abstract

Background: Increased antibiotic resistance is one of the major factors contributing to the failure of *H. pylori* eradication. This study aimed to compare the efficacy and safety of doxycycline and amoxicillin, both critical components for bismuth-based quadruple therapy, for the first-line treatment of *H. pylori*-infected duodenal ulcers. Methods: An open, randomized case-controlled, multicenter trial was conducted in seven hospitals in China. A total of 184 eligible participants were divided into an IDFB (ilaprazole 5 mg, doxycycline 100 mg, furazolidone 100 mg, and bismuth 220 mg bid) or IAFB (ilaprazole 5 mg, amoxicillin 1000 mg, furazolidone 100 mg, and bismuth 220 mg bid) group for 14 days. Both groups were administrated with ilaprazole 5 mg qd for another 14 days. The main outcome was an *H. pylori* eradication rate; secondary outcomes were ulcer healing, relief of symptoms, and incidence of adverse effects. Results: The *H. pylori* eradication rates were 85.9% (95% CI 78.6–93.9) in the IDFB vs. 84.8% (95% CI 77.3–92.3) in the IAFB group in ITT analysis (*p* > 0.05), and 92.9% (95% CI 87.4–98.5) vs. and 91.8% (95% CI 85.8–97.7) in PP analysis (*p* > 0.05). The overall ulcer healing rates of IDFB and IAFB were 79.1% and 84.7% (*p* > 0.05), both effective in relieving symptoms. Only nine participants had adverse reactions in this trial (4/92 in IDFB and 5/92 in IAFB). Conclusion: A bismuth quadruple regimen containing doxycycline or amoxicillin could be an effective and safe treatment for *H. pylori* eradication, while doxycycline replacement is an alternative for participants with penicillin allergy.

## 1. Introduction

Although the detection and treatment of *Helicobacter pylori* (*H. pylori*) are more accessible, *H. pylori* infection is highly prevalent worldwide, especially in developing countries [1]. Peptic ulcers, such as duodenal ulcers, are one of the most common diseases caused by *H. pylori* infection [2]. It is likely to cause delayed healing and complications such as bleeding and perforation and may even endanger the patient’s life once not well treated [3]. The treatment of *H. pylori* is low-cost and highly effective. Authors of consensus guidelines strongly recommend that timely eradication therapy is necessary for *H. pylori*-infected ulcer participants [4,5,6,7].

Increased antimicrobial resistance with high resistance rates to clarithromycin, levofloxacin, and metronidazole has been found [8], which is the most common reason the bismuth-based quadruple therapy (BQT) approach fails [4,6]. In contact, antibiotics such as furazolidone, amoxicillin, and tetracycline have comparatively low resistance and can achieve effective eradication, especially when used in BQT [9]. As the most used semi-synthetic penicillin broad-spectrum-lactam antibiotic, amoxicillin consistently demonstrates good efficacy and compliance both in the initial and remedial eradication treatment of *H. pylori*. However, penicillin is one of the most common drugs to cause an allergic reaction with an allergy rate of 5–10% of the population [10,11], patients treated with amoxicillin are prone to erythema as well as severe adverse events such as hemorrhagic enteritis [12]. The selection of a suitable therapeutic regimen is thus difficult for *H. pylori*-infected participants with penicillin allergies.

A series of previous studies have tended to recommend proton pump inhibitor (PPI)-clarithromycin-metronidazole and/or bismuth-containing regimens for first-line eradication of *H. pylori* infection with penicillin allergy, but such eradication regimens are less effective due to the high drug resistance of clarithromycin and metronidazole in most regions, achieving only 50–70% eradication rates [13,14]. The use of tetracycline instead of clarithromycin was slightly more effective [15]. While tetracycline is the recommended drug of conventional guidelines for eradicating *H. pylori* [16,17], others have shown that the use of tetracycline increases the risk of side reactions [18], limiting its clinical use. Doxycycline is a second-generation semi-synthetic tetracycline antibiotic that is less commonly used in *H. pylori* eradication therapies [19]. Interestingly, its range of medications is similar to that of amoxicillin, which could theoretically replace amoxicillin for *H. pylori* eradication treatment.

In this parallel opened, randomized controlled, multicenter study, the efficacy and safety of bismuth-based quadruple therapy containing doxycycline or amoxicillin for initially treating participants with *H. pylori* infection-associated duodenal ulcer were compared. The overall aim was to provide a safe, effective new treatment for the eradication of *H. pylori* in penicillin-allergic patients.

## 2. Methods

### 2.1. Study Design and Patients

A parallel opened, randomized controlled, multicenter trial was conducted in seven tertiary class hospitals in five cities of Hunan Province, China, between November 2017 and September 2019. A random sequence was constructed in 1:1, which was divided into two groups. Outpatients were randomly assigned to one of the groups in sequence. The study terminated at the last expiration date of the same batch of drugs. This study was approved by the committee of central research institution (Third Xiangya Hospital of Central South University), then other hospitals recorded and followed this approval. The study also has been registered on ClinicalTrials.gov (number NCT03342456). All participants gave their informed consent.

Participants all had a gastroscopic diagnosis of an active duodenal ulcer (Sakita and Miwa ulcer stage A1/A2) with a diameter of 0.3–2.0 cm and counts of less than two, accompanied by gastrointestinal symptoms. Further, participants ranged in age from 18 to 65, and *H. pylori* infection was initially found through 13C/14C-urea breath test or histopathological examination. Potential participants who had taken PPI in the past two weeks, antibiotics and/or bismuth in the past four weeks, or had been taking non-steroidal anti-inflammatory drugs and adrenocortical hormones, or were allergic to the tested drugs were excluded. Patients who were suspected to have cancerous ulcers or who had tumor alarm symptoms, other severe diseases of the digestive and other systems, or surgical history were also excluded. Potential participants who were pregnant, breastfeeding, or planning a family, who had severe heart and lung diseases, or who abused drugs and/or alcohol were unsuitable for this clinical drug trial.

### 2.2. Interventions

Participants were randomly assigned to one of two groups by computer-generated random sequences. Those in the IDFB group received 5 mg of ilaprazole (Livzon Pharmaceutical Group Inc., Zhuhai, China, 100 mg of doxycycline (Yung Shin Pharm, Ind., Kunshan Co., Ltd., Kunshan, China), 100 mg of furazolidone (Yunpeng Shanxi Pharmaceutical Co., Ltd., Linfen, China), and 220 mg of bismuth potassium bismuth citrate (Livzon Pharmaceutical Group Inc., Zhuhai, China) twice a day for 14 days. The IAFB group was treated with 1000 mg of amoxicillin (Zhuhai United Laboratories (Zhongshan) Co., Ltd., Zhongshan, China) instead of the doxycycline. After 14 days, 5 mg of ilaprazole was given to both groups, once a day, for an additional 14 days.

### 2.3. CurativeEffects Evaluation

*H. pylori* eradication: A negative 13C/14C-urea breath test four weeks after discontinuation of all drugs indicated successful eradication of *H. pylori*. The *H. pylori* eradication rate = number of cases eradicated/total number of cases ×100%.

Therapeutic effects of duodenal ulcer: Four weeks after eradication drugs were taken, the gastroscopic manifestations were compared with those during screening gastroscopy. The degree of ulcer healing was assessed using the classification of Sakita and Miwa [20]. The criterion for cure efficiency was whether the ulcer disappeared or was in Stage S1/S2 [21]. Overall cure rate = number of cured cases/total number of cases ×100%.

Evaluation of symptomatic remissions: All symptoms including epigastric pain, heartburn, acid regurgitation, nausea and vomiting, belching, and abdominal distension were recorded before and on days 14 and 28 after taking medicine. Severity was graded as 0 (none), 1 (mild: easy to tolerate), 2 (moderate: affecting normal life), or 3 (severe: unable to live a normal life) [22]. The method has been verified and used to assess gastrointestinal symptoms in Chinese patients [23,24].

### 2.4. Outcome Evaluation

The primary outcome was the *H. pylori* eradication rate, while the therapeutic effects on the duodenal ulcer and symptomatic remissions were the secondary outcomes.

### 2.5. Side Effects Evaluation

Side effects including abdominal pain, diarrhea, erythema, nausea, and dizziness were recorded on days 14 and 28, as well as 28 days after withdrawal of the drugs. Any necessary treatment was given.

### 2.6. Sample Size Estimation and Statistical Analysis

A non-inferiority design was used in this study. Based on previous studies, we assumed that the eradication rate of the amoxicillin-containing quadruple eradication program was 90% [25,26] and 92% for the doxycycline regimen [27]. According to PASS12, the non-inferiority margin should be set to −10%, α = 0.025 and β = 0.20, and each group should include at least 80 individuals. In this trial, 200 participants were initially included during screening.

*H. pylori* eradication rates and gastroscopic cure rates for ulcers were expressed as percentages and evaluated with 95% confidence intervals. The *H. pylori* eradication rate was analyzed by per-protocol (PP) and intention-to-treat (ITT), while symptom relief and ulcer effectiveness were analyzed by PP analysis alone. *p* < 0.05 represents statistical significance. All data were analyzed using SPSS18. A chi-square test or Fisher’s exact test was used to assess the significance of the categorical data, while a t-test was used for continuous variables.

## 3. Results

A total of 200 outpatients diagnosed with duodenal ulcers and *H. pylori* infection were screened in different hospitals. Sixteen patients were excluded for the following reasons: history of taking PPI and other drugs in the past two weeks (*n* = 5); ulcer not in active phase (*n* = 4); ulcer with active bleeding (*n* = 3); and the refusal of the treatment regimen (*n* = 4). Following this, 184 eligible participants were randomly assigned to two groups: the IDFB group (92 participants) and the IAFB group (92 participants). Throughout the study, 14 participants (7.6%) dropped out, of which 12 were lost to follow-up as they could not be contacted or refused to recheck, and two discontinued because of giving up taking medication by themselves (Figure 1).

**Figure 1 pathogens-11-01549-f001:**
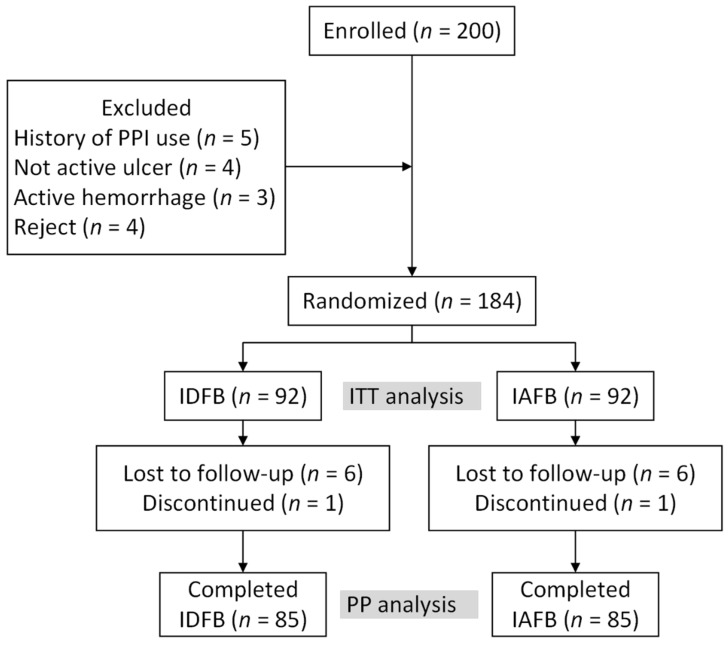
Study flow chart. Baseline data concerning age, sex, vital signs, and history of the study population were analyzed. No statistical difference was found between the two groups (*p* > 0.05, Table 1).

The eradication rates in the ITT analysis were 85.9% (95% CI 78.6–93.9) for the IDFB group and 84.8% (95% CI 77.3–92.3) for the IAFB group, *p* = 0.84. The use of PP analysis showed that eradication rates for the two groups were 92.9% (95% CI 87.4–98.5) and 91.8% (95% CI 85.8–97.7), respectively, *p* = 0.77. No statistical difference in the eradication rate between the two groups was found for either analysis (Table 2).

Ulcer healing was determined by comparing endoscopic features of participants who had re-examined gastroscopy before and after taking medicine. The ulcer cure rate for the two groups was 79.1% (95% CI 69.1–89.1) and 84.7% (95% CI 76.2–93.2) by PP analysis, with no significant difference found between the two groups (*p* = 0.39, Table 3).

Subjective symptoms were quantified by a rating scale. Two groups presented similar scores of the baseline symptoms (Table 4). Both groups achieved good overall symptom remission after participating in the study. The decreased total scores were 4.65 ± 0.32 in IDFB and 5.09 ± 0.34 in IAFB after 14 days and 5.25 ± 0.37 and 5.73 ± 0.33 after 28 days of therapies, respectively. Two groups reduced more than 96% of the initial symptom scores (Table 5).

A total of nine participants had adverse reactions during the trial. These included abdominal pain, diarrhea, erythema, nausea, and dizziness, all of which were mild with spontaneous remission. The incidence of both side effects was low, with four (4.3%) in the IDFB group and five (5.4%) in the IAFB group. Similarly, no statistical significance was found between the two groups (Table 6).

## 4. Discussion

In this study, participants with *H. pylori* infections complicated with duodenal ulcer were treated for the first-time using doxycycline or amoxicillin combined with furazolidone and Ilaprazole as a 14-day bismuth agent quadruple therapy with significant, remarkable results. Both therapeutic schemes achieved good eradication efficacy, a clearly deceased ulcer area, effectively alleviated gastric symptoms, and had few side effects. Additionally, these results did not differ significantly between the two groups, suggesting that doxycycline could replace amoxicillin as a new option for *H. pylori* eradication.

Due to widespread, inappropriate antibiotic use, antimicrobial resistance is a common factor affecting treatment to eradicate *H. pylori* [28]. Raising eradication efficiency can be achieved by selecting better and large doses of PPI, and adding bismuth, an antibiotic combination, or personalized medication following a drug susceptibility test [29]. Unfortunately, susceptibility testing for patients is rarely performed and unrealistic in China, because of our large *H. pylori* infected population and the limitation of testing technology in primary hospitals. Compared with antimicrobial susceptibility testing, an empiric 14-day bismuth-containing quadruple therapy is an option that can improve the eradication effect distinctly with a higher cost–benefit ratio [6]. Our research is further evidence of the superiority of such a scheme; both types of eradication therapy had high cure rates of *H. pylori* infection in PP analyzed. Moreover, Ilaprazole was selected as PPI in the current study. It is the third generation of PPI and not metabolized by CYP2C19 enzymes, which has a prolonged plasma half-life hence small individual differences [30]. Ilaprazole can effectively inhibit gastric acid secretion with a low dose and fewer times of administration [31,32]. As well as amoxicillin and doxycycline, furazolidone is another antibiotic used in this study, which is a nitrofuran antibiotic with low resistance rates [33]. These drugs also helped the trial run smoothly, and effectively alleviate the symptoms of ulcer participants and promote ulcer healing

Authors of previous studies have shown that the total eradication rate of *H. pylori* can reach approximately 90.0% with a combination of two antibiotics containing amoxicillin [9]. As well as amoxicillin and doxycycline, furazolidone is another antibiotic used in this study, which is a nitrofuran antibiotic with low resistance rates [33]. Quadruple therapies based on a combination of furazolidone and amoxicillin were effective in eradicating *H. pylori*, with a rate of more than 90% at seven days treatment to 10 days [34]. In this trial, the eradication rate of the control group (IAFB) was as high as 91.8%, which is similar to the efficacy found in previous studies. However, amoxicillin was quite effective in traditional therapies, including the IAFB scheme in this study. The allergic phenomenon of penicillin is still a challenge for *H. pylori* eradication. Thus, many authors have focused on anti-*H. pylori* therapy for participants who have an amoxicillin allergy [35]. In addition to conventional alternatives for clarithromycin and metronidazole, previous retrospective research has demonstrated that a first-line bismuth-based quadruple therapy (tetracycline and metronidazole or furazolidone) received an eradication rate of over 91%, while about 30% were adverse events [36,37]. Recent first-line studies showed non-inferiority in the eradication efficiency between tetracycline and amoxicillin bismuth-based quadruple therapies, but the tetracycline-containing regimens showed a higher risk of adverse reactions [17,38]. These data indicated that traditional tetracycline was effective in *H. pylori* eradication treatments and, particularly, could be used instead of amoxicillin. However, at the same time, it was associated with a greater risk of side effects that led to its limited clinical application.

As a second-generation, semi-synthetic tetracycline antibiotic, doxycycline has approximately the same antibacterial spectrum as tetracycline, but a higher bioavailability [39]. A study between doxycycline and tetracycline for a 10-day regimen found that the doxycycline group not only had a higher eradication rate but also far fewer side effects (11.6%) than the tetracycline group (31.0%) [40]. Thus, doxycycline appears to show better safety by comparison. In the current study, only 4.6% of the participants in the IDFB group experienced mild adverse reactions that disappeared with the cessation of the drug, further verifying the safety of doxycycline. In addition, a meta-analysis found that treatment with doxycycline was more effective for eradicating *H. pylori* than those without semisynthetic tetracycline [41]. Researchers have found that a 14-day doxycycline-containing (100 mg bid) bismuth-basted quadruple regimen could gain approximately 93.8% eradication in the first-line treatment [42] but was not as satisfactory in a 5-day regimen [43]. Therefore, doxycycline seems more effective when used in an entire course with a bismuth-containing quadruple regimen. From our results, the therapeutic effect in the doxycycline and furazolidone group (IDFB) was up to 92.9%, slightly higher than the amoxicillin therapy, suggesting a potential application value of doxycycline for the treatment of *H. pylori* eradication. In addition, this trial has reflected the superior performance of doxycycline in symptom relief for patients with duodenal ulcers, which other researchers rarely demonstrate.

Doxycycline has not been widely used for eradicating *H. pylori*, meaning there are limited studies in this area so far [44,45]. There is controversy about the application of its course of treatment, dosage, and other details. In this prospective study, doxycycline showed significant efficacy and safety, possibly due to the dose, the 14-day full course, the low resistance of antibiotic combination, and the use of bismuth. It is suggested that the use of doxycycline in the first-line treatment for *H. pylori* infection could be expanded, especially in antibacterial treatment for patients allergic to penicillin. However, there are some limitations to this study. We did not test patients’ sensitivity to each medicine nor did we explore whether a shorter duration of antibiotic treatment (such as 7 or 10 days) could achieve similar efficacy. This can be clarified in future in-depth studies.

## 5. Conclusions

To conclude, doxycycline-containing bismuth-based quadruple therapy is safe and effective for the eradication treatment of *H. pylori* infection, providing a good alternative for amoxicillin-allergic patients.

## Figures and Tables

**Table 1 pathogens-11-01549-t001:** Demographic analysis of the study population.

Variable	IDFB (*n* = 92)	IAFB (*n* = 92)	*p* Value
Age (y)	41.84 ± 11.48	43.09 ± 11.31	0.46 ^a^
Gender (n,%)			
Male Female	56 (60.9%) 36 (39.1%)	61 (66.3%) 31 (33.7%)	0.44 ^b^
BMI	22.53 ± 2.61	22.29 ± 3.00	0.56 ^a^
Vital Signs			
ortho-arteriotomy (n,%)	87 (94.6%)	86 (93.5%)	0.76 ^b^
respiratory rate (t)	16.52 ± 2.08	16.73 ± 2.31	0.53 ^a^
pulse rate (b/min)	75.48 ± 8.30	73.96 ± 7.21	0.19 ^a^
temperature (℃)	36.79 ± 1.04	36.67 ± 0.30	0.31 ^a^
Course (d)	51.77 ± 130.55	39.49 ± 84.36	0.45 ^a^
Past-history			
none (n,%)	78 (84.8%)	83 (90.2%)	0.27 ^b^

^a^*t*-test; ^b^ Chi-square test; BMI, body mass index.

**Table 2 pathogens-11-01549-t002:** Eradication of *H. pylori* infection in different treatment groups.

Efficacy	IDFB (*n* = 92)	IAFB (*n* = 92)	*χ* ^2^	*p* Value ^b^
Eradication success	79	78		
Eradication failure	6	7		
Cases dropped	7	7		
Eradication rate				
TT (95%CI)	85.9% (78.6–93.9)	84.8% (77.3–92.3)	0.04	0.84
PP (95%CI)	92.9% (87.4–98.5)	91.8% (85.8–97.7)	0.08	0.77

^b^ Chi-square test; ITT, intention-to-treat; PP, per-protocol; CI, confidence interval.

**Table 3 pathogens-11-01549-t003:** Therapeutic effects of duodenal ulcer in different treatment groups.

Efficacy	IDFB (*n* = 67)	IAFB (*n* = 72)	*χ* ^2^	*p* Value ^b^
Obvious effect	53	61		
No effect	14	11		
Effective rate	79.1% (69.1–89.1)	84.7% (76.2–93.2)	0.74	0.39
P (95%CI)				

^b^ Chi-square test; PP, per-protocol; CI, confidence interval.

**Table 4 pathogens-11-01549-t004:** Baseline of clinical symptom scores of two groups.

Variable	IDFB (*n* = 85)	IAFB (*n* = 85)	*p* Value ^a^
Epigastric pain			
day	1.13 ± 0.07	1.27 ± 0.07	0.17
night	1.08 ± 0.09	1.28 ± 0.09	0.13
Heartburn			
day	0.35 ± 0.06	0.45 ± 0.07	0.32
night	0.28 ± 0.06	0.35 ± 0.07	0.42
Acid regurgitation			
day	0.53 ± 0.08	0.51 ± 0.08	0.84
night	0.33 ± 0.07	0.29 ± 0.06	0.71
Nausea and vomiting	0.33 ± 0.06	0.33 ± 0.07	1
Belching	0.65 ± 0.08	0.65 ± 0.07	1
Abdominal distension	0.75 ± 0.08	0.82 ± 0.08	0.53
Total	5.44 ± 0.37	5.95 ± 0.33	0.29

^a^ *t*-test.

**Table 5 pathogens-11-01549-t005:** Changes of clinical symptom scores after 14 and 28 days of therapy in different groups.

Variable	14 Days			28 Days		
	IDFB (*n* = 85)	IAFB (*n* = 85)	*p* Value ^a^	IDFB (*n* = 85)	IAFB (*n* = 85)	*p* Value ^a^
Epigastric pain						
day	1.00 ± 0.07	1.05 ± 0.08	0.67	1.11 ± 0.07	1.24 ± 0.08	0.25
night	0.93 ± 0.09	1.12 ± 0.10	0.16	1.05 ± 0.09	1.26 ± 0.09	0.14
Heartburn						
day	0.27 ± 0.06	0.41 ± 0.07	0.12	0.32 ± 0.06	0.45 ± 0.07	0.18
night	0.22 ± 0.05	0.33 ± 0.07	0.2	0.25 ± 0.06	0.35 ± 0.07	0.31
Acid regurgitation						
day	0.47 ± 0.08	0.46 ± 0.08	0.92	0.52 ± 0.08	0.51 ± 0.08	0.91
night	0.31 ± 0.06	0.26 ± 0.07	0.61	0.32 ± 0.07	0.28 ± 0.06	0.64
Nausea and vomiting	0.28 ± 0.06	0.29 ± 0.07	0.9	0.33 ± 0.06	0.32 ± 0.07	0.96
Belching	0.55 ± 0.07	0.44 ± 0.06	0.22	0.62 ± 0.08	0.51 ± 0.07	0.37
Abdominal distension	0.61 ± 0.07	0.74 ± 0.08	0.30	0.74 ± 0.08	0.82 ± 0.08	0.43
Total	4.65 ± 0.32	5.09 ± 0.34	0.34	5.25 ± 0.37	5.73 ± 0.33	0.33

^a^ *t*-test.

**Table 6 pathogens-11-01549-t006:** Side effects associated with the two groups (n, %).

Side Effects	IDFB (*n* = 92)	IAFB (*n* = 92)	*p* Value ^c^
Abdominal pain	1		
Diarrhea	1	2	
Dizziness	1	1	
Nausea	1	1	
Erythema		1	
Total	4 (4.3%)	5 (5.4%)	1

^c^ Fisher’s exact test.

## Data Availability

Not applicable.

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
