# Peer review of "A Comparison of Doxycycline and Amoxicillin Containing Quadruple Eradication Therapy for Treating Helicobacter pylori-Infected Duodenal Ulcers: A Multicenter, Opened, Randomized Controlled Trial in China"

_pathogens, 2022, doi:10.3390/pathogens11121549_

Round 1

Reviewer 1 Report

This is an interesting manuscript regarding alternative drug treatment for Hp infection. However, it raises some concerns.

L. 55, 179 and Tab. 6: what is "erythra"?

L. 81-82: the authors report that "This study was all approved by the ethics committees of these seven hospital", but in the ethical statement at the end of the manuscript is states that only one hospital approved the study. The authors should report all the IECs approvals and the relatedprotocol numbers before publishing approval.

Fig. 1 reports 14 missed patients (lost or discontinued) but in the text are not evident the reasons of such abandon.

Tab. 1 must be revised reporting females number and percentages, and escluding non-demographic data (Vital Signs, BMI) and adding data regarding study and occupational status of the enrolled patients.

The text must be revised for typos.

Author Response

Dear reviewer,

 We feel great thanks for your professional review work on our article. As you are concerned, several problems need to be addressed. We have made many corrections to our previous draft according to your nice suggestions,  the detailed modifications are listed attached. Please see the attachment.

Reviewer 2 Report

This publication describes the results of a clinical trial for treatment of H. pylori infection using either doxycycline or amoxicillin in the quadruple therapy. The paper is well written but there are minor things to consider;

The title should be "A comparison of  doxycycline or amoxicillin containing quadruple eradication therapies for treating Helicobacter pylori infected duodenal ulcers: A multicenter, opened, randomized controlled trial in China"

Line 38, Helicobacter pylori is not italicised 

Results start with 200 patients and 19 were excluded (Line 142), but the total participants described in the study are 184. the numbers do not matchup

All the data is presented in tables, that are informative but not very reader friendly. Some of the data can be presented in form of charts e.g. Table 2 and Table 3

Author Response

Dear reviewer,

Thank you for your positive comments and valuable suggestions to improve the quality of our manuscript. We have carefully considered all your comments and revised our manuscript accordingly. The detailed point-by-point responses are attached below. Please see the attachment.

Reviewer 3 Report

The manuscript enriches the big number of studies concerning the eradication of H. pylori., My comments are:

1. Some sentences/term are rather difficult to understand: what is "normative radical treatment"? (surgery?) What does mean the term "erythra"? (erythema?)

2. Some pharmacologic  data about ilaprozole, a rather unknown PPI for the non-Chinese rweaders, should be welcome.

3. Geographic differences of eradication rates with doxycycline must be included and commented

4. Is the GSRS scale translated and valiudatwed in Chinese - this must be mentioned

5. Reference 7 must be replaced in the text and list because in the meantime, the Maastricht VI/Florence consensus was published

Author Response

Dear reviewer,

Thank you for your nice comments on our article. According to your suggestions, we have supplemented several details here and corrected several mistakes in our previous draft. Based on your comments, we also attached a point-by-point letter to you. Please see the attachment.

Round 2

Reviewer 1 Report

The authors addressed all the comments and their replies are satisfactory.